# Marine-Derived Sulfated Glycans Inhibit the Interaction of Heparin with Adhesion Proteins of *Mycoplasma pneumoniae*

**DOI:** 10.3390/md22050232

**Published:** 2024-05-20

**Authors:** Jiyuan Yang, Yuefan Song, Ke Xia, Vitor H. Pomin, Chunyu Wang, Mingqiang Qiao, Robert J. Linhardt, Jonathan S. Dordick, Fuming Zhang

**Affiliations:** 1The Key Laboratory of Molecular Microbiology and Technology, Ministry of Education, College of Life Sciences, Nankai University, Tianjin 300071, China; yangj25@rpi.edu (J.Y.); qiaomq@nankai.edu.cn (M.Q.); 2Department of Chemistry and Chemical Biology, Center for Biotechnology and Interdisciplinary Studies, Rensselaer Polytechnic Institute, Troy, NY 12180, USA; songy11@rpi.edu (Y.S.); xiak@rpi.edu (K.X.); wangc5@rpi.edu (C.W.); linhar@rpi.edu (R.J.L.); 3Department of BioMolecular Sciences, Research Institute of Pharmaceutical Sciences, The University of Mississippi, Oxford, MS 38677, USA; vpomin@olemiss.edu; 4Department of Chemical and Biological Engineering, Rensselaer Polytechnic Institute, Troy, NY 12180, USA

**Keywords:** *Mycoplasma pneumoniae*, HSPGs, heparin, marine sulfated glycans, surface plasmon resonance

## Abstract

*Mycoplasma pneumoniae*, a notable pathogen behind respiratory infections, employs specialized proteins to adhere to the respiratory epithelium, an essential process for initiating infection. The role of glycosaminoglycans, especially heparan sulfate, is critical in facilitating pathogen–host interactions, presenting a strategic target for therapeutic intervention. In this study, we assembled a glycan library comprising heparin, its oligosaccharide derivatives, and a variety of marine-derived sulfated glycans to screen the potential inhibitors for the pathogen–host interactions. By using Surface Plasmon Resonance spectroscopy, we evaluated the library’s efficacy in inhibiting the interaction between *M. pneumoniae* adhesion proteins and heparin. Our findings offer a promising avenue for developing novel therapeutic strategies against *M. pneumoniae* infections.

## 1. Introduction

*Mycoplasma pneumoniae* represents a significant etiological agent behind community-acquired respiratory infections, manifesting a spectrum from mild tracheobronchitis to severe pneumonia, predominantly in pediatric and young adult populations [1,2]. Transmission of this pathogen predominantly occurs through respiratory droplets in environments (airborne) characterized by high density, poor ventilation, or close contact, with an incubation period ranging from one to three weeks [3]. *M. pneumoniae* infections exhibit a year-round global presence, subject to periodic epidemics attributed to factors such as diminished herd immunity and the emergence of novel subtypes, with notable outbreaks recorded in late 2019 and early 2020 across multiple nations, mainly in Europe and Asia [4,5]. Reports indicate substantial occurrences of co-infection involving *M. pneumoniae* alongside various bacteria and viruses, with certain instances leading to severe clinical outcomes [6,7]. Amidst the COVID-19 pandemic, non-pharmaceutical interventions have markedly influenced the epidemiological landscape of *M. pneumoniae* infections. A global survey spanning April 2020 to March 2021 revealed a significant decline in *M. pneumoniae* incidence, attributed to enhanced hygiene, social distancing, and lockdown measures. However, the pandemic’s aftermath has seen a resurgence in certain regions, accompanied by an increase in macrolide-resistant strains and co-infections [6,8]. This situation underscores the need for ongoing epidemiological monitoring and the adjustment of treatment approaches accordingly.

The pathogenesis of *M. pneumoniae* is closely associated with its ability to adhere to the respiratory epithelium, a critical step for colonization and subsequent infection. This bacterium employs a specialized terminal attachment organelle [9], alongside pathogenic factors that facilitate adhesion and subsequent cellular and structural alterations within host cells. *M. pneumoniae*’s adhesion and motility on host cells are critically influenced by the interaction with host receptors, predominantly sialylated and sulfated oligosaccharides on the cell surface (glycocalyx) [10]. The composition and concentration of these cellular receptors significantly modulate *M. pneumoniae*’s adhesion capacity, thereby influencing disease progression and outcomes [11]. The bacterium’s tip terminal structure, harboring an assembly of proteins including the primary surface adhesins P1 and P30, alongside auxiliary proteins P116, HMW1-HMW5, and the additional components P40 and P90, facilitates this adherence process [12,13,14]. Among the proteins involved in this adherence interaction, P1 and P30 are recognized as key adhesins. The membrane-associated P1 protein is instrumental in mediating bacterial attachment to host cells, further enabling the bacterium’s gliding motility across the cellular surface [15]. Additionally, the immunogenic properties of the P1 protein’s carboxyl terminus have been identified as potential diagnostic and pathophysiological research targets [16]. P30 is a membrane protein located at the distal tip of the terminal organelle that shares sequence similarities with particular domains with the P1 protein and plays a crucial role in cytadherence and gliding motility [17,18]. Given their critical roles in *M. pneumoniae* pathogenesis, targeting the interaction between P1, P30, and host cell receptors presents a promising therapeutic strategy.

Glycosaminoglycans (GAGs), a family of linear, negatively charged polysaccharides, play a pivotal role in cell surface and extracellular matrix interactions, significantly impacting cellular functions and pathogen–host interactions. Among these, heparan sulfate (HS) emerges as a critical component within the mammalian system, predominantly through its incorporation into HS proteoglycans (HSPGs). These complexes are integral to various cellular regulatory mechanisms, including receptor activation, signaling, cytoskeletal dynamics, and cellular communication [19]. Notably, HSPGs have been identified as key facilitators in the invasion of host cells by a wide array of pathogens, including viruses such as coronaviruses and retroviruses, highlighting their role in microbial pathogenesis [20]. Research indicates a significant role of GAGs in mediating the adherence and invasion of host cells by pathogens, including *Mycoplasma hyopneumoniae* [21] and *Chlamydia pneumoniae* [22]. These insights suggest that further investigation into the specific interactions between *M. pneumoniae* adherence proteins and GAGs could reveal significant aspects of the pathogen’s adherence mechanism and potentially lead to new approaches for preventing *M. pneumoniae* infections. Furthermore, natural sulfated glycans from marine organisms exhibit unique sulfation patterns and molecular structures, distinct from those found in terrestrial sources [23]. These structural variances confer diverse biological properties and interactions, making marine-derived glycans a valuable resource for novel biomedical applications. Marine sulfated polysaccharides have been demonstrated to possess potential anticoagulant, antitumor, and antiviral activities [24]. With increasing emphasis on sustainability, marine sources provide a viable alternative to terrestrial sources for harvesting biologically active compounds, and they also offer promising targets for exploring novel anti-adherence therapies.

In this study, a glycan library (Figure 1) comprising heparin, its derived oligosaccharides, and marine sulfated glycans of various structures was prepared to assess the potential inhibitory effects on *M. pneumoniae* adherence proteins’ interaction with heparin. Inhibitory activity was investigated by Surface Plasmon Resonance (SPR) spectroscopy. Our findings revealed a significant inhibition of heparin binding to the P1 C-terminal (P1-C) and P30 proteins in the presence of selected sulfated glycans. This inhibition underscores the therapeutic promise of sulfated glycans in disrupting critical pathogen–host interactions mediated by adherence proteins.

## 2. Results and Discussion

### 2.1. Binding Kinetics and Affinity of the Interaction between Heparin and M. pneumoniae Proteins

Adherence represents the initial step in *M. pneumoniae*’s colonization of the respiratory epithelium, precipitating subsequent infection. The P1 protein plays a crucial role in the pathogenesis of infection, distinguished by its significant immunogenicity and antigenic specificity [25]. Specifically, the C-terminal region of P1 (P1-C) plays a crucial role in binding to various host cell molecules, enhancing its utility for immunodiagnostics due to superior sensitivity [15,26]. The P30 protein is similarly critical for cytadherence, gliding motility, and cell division [27]. As a vital element of *M. pneumoniae*’s terminal organelle, P30 has undergone cloning and detailed characterization, highlighting its utility as a diagnostic antigen [28]. This positions it alongside the P1 protein as a valuable tool for immunodiagnostics of *M. pneumoniae* infections. HSPGs play a critical role in a wide range of cellular regulatory mechanisms, with one of their key functions being the facilitation of pathogenic invasion into host cells. Heparin, due to its structural similarity to HS found on cell surfaces, consistently serves as a model for studying GAG–protein interactions.

To investigate the binding affinity of *M. pneumoniae* adhesion proteins to heparin, we utilized SPR with a heparin-coated chip, analyzing the binding kinetics of P30 and P1-C to heparin. The binding kinetics and affinity were determined by globally fitting the complete association and dissociation phases using a 1:1 Langmuir binding model. Sensorgrams revealed the interaction dynamics between these proteins and heparin (Figure 2), with kinetic parameters detailed in Table 1. These parameters include the association rate constant (ka), the dissociation rate constant (kd), and the equilibrium dissociation constant (*K_D_*, calculated as kd/ka). The *K_D_* values for P30 and P1-C were 80.3 nM and 16.4 nM, respectively, indicating a significant binding affinity of both *M. pneumoniae* proteins for heparin. These proteins demonstrated a dose-dependent binding to heparin, characterized by slow association and dissociation rates. Notably, the P1-C protein exhibited a stronger binding affinity for heparin, underscoring its potential importance in heparin interaction studies.

### 2.2. SPR Solution Competition between Surface-Immobilized Heparin and Heparin Oligosaccharides and Desulfated Heparins

GAGs are an essential class of macromolecules characterized by their long, linear chains of negatively charged polysaccharides. Heparin and HS consist of disaccharide repeating units composed of uronic acid and glucosamine and have a structure of [→4)α-L-iduronic/β-D-glucuronic acid (IdoA/GlcA)(1→4)α-D-glucosamine (GlcN)(1→], featuring varying sulfation substitutions at the 2-*O*-position of uronic acid and the 3-*O*, 6-*O*, *N*-positions of glucosamine and/or *N*-acetylation at the glucosamine [29]. To explore the impact of oligosaccharide chain length on the binding interactions between heparin and *M. pneumoniae* adherence proteins, we conducted solution/surface competition SPR experiments. These assessed the inhibitory potential of various heparin oligosaccharides with differing polymerization degrees on the interactions between surface-bound heparin and proteins P30 and P1-C. Solutions of P30 protein (500 nM) and P1-C protein (100 nM) were each pre-mixed with heparin oligosaccharides (1000 nM). The solution competition analysis between heparin and its oligosaccharides revealed significant inhibitory effects on the binding of the P30 protein to the heparin-coated surface, as shown in Figure 3A. Specifically, heparin itself presented a 76% inhibition of P30 protein binding, while heparin oligosaccharides, spanning degrees of polymerization (dp) from dp4 to dp18, showed variable binding inhibitions ranging from 20% to 52%. In the case of the P1-C protein, heparin demonstrated an 89% reduction in binding to the surface-immobilized heparin, with oligosaccharides from dp4 to dp18 inhibiting binding by 17% to 35% (Figure 3C,D). Notably, for both *M. pneumoniae* proteins, no discernible correlation between glycan chain length and the extent of binding inhibition was observed.

Moreover, we investigated the structural factors influencing the competitive binding of heparin by evaluating the inhibitory effects of chemically desulfated heparins (maintaining comparable chain lengths) on the interaction between *M. pneumoniae* proteins and surface-immobilized heparin. All variants of desulfated heparin (2-DeS, 6-DeS, and *N*-DeS heparin) demonstrated reduced binding affinity to the immobilized heparin in comparison to the control (Figure 3E–H), suggesting that the removal of sulfate groups diminishes binding. However, no statistically significant differences among the desulfated heparin types were observed, indicating that the specificity of binding does not depend on the position of the sulfate groups but rather on the presence of adequate charge.

### 2.3. Inhibition of Isostichopusbadionotus-Sourced Sulfated Glycans on the Interaction between Heparin and M. pneumoniae Proteins

IbSF and IbSFucCS, sulfated glycans extracted from the sea cucumber *Isostichopus badionotus* (Ib), possess distinct structural motifs: IbSF is characterized by the repeating unit [→3)-α-Fuc2,4S-(1→3)-α-Fuc2S-(1→3)-α-Fuc2S-(1→3)-α-Fuc-(1→], while IbFucCS comprises [→3)-β-GalNAc4,6S-(1→4)-β-GlcA[(3→1)Y]-(1→], where Y = α-Fuc2,4S (96%) or α-Fuc4S (4%) [30,31] (Figure 1). These compounds have demonstrated significant anticoagulant and antithrombotic activities, alongside potent inhibitory effects against various coronavirus strains, including the wild-type and Delta variants of SARS-CoV-2 and MERS as detailed in our prior research [32].

In this study, we explored the inhibitory potential of IbSF, IbFucCS, and their desulfated derivatives (desIbSF and desIbFucCS) on the interaction between heparin and *M. pneumoniae* proteins P30 and P1-C using solution/surface competition assays. Each glycan, at a consistent concentration of 10 μg/mL, was pre-mixed with solutions of P30 (500 nM) and P1-C (100 nM), respectively, before the injection. The results indicated a pronounced inhibition of the binding between P30 and P1-C proteins to surface-immobilized heparin by all Ib glycans (Figure 4). However, upon complete desulfation, both desIbSF and desIbFucCS exhibited diminished efficacy in competing with heparin for binding to the *M. pneumoniae* proteins. These findings suggest the potential of utilizing GAGs or their mimetics as therapeutic agents against *M. pneumoniae* infection by competitively blocking pathogen adherence to host cells. Crucially, sulfation appears as a key structural feature for the interaction of marine-derived sulfated glycans with *M. pneumoniae* proteins.

### 2.4. Inhibition of Glycans from Holothuria floridana, Lytechinus variegatus and Pentacta pygmaea on the Interaction between Heparin and M. pneumoniae Proteins

Sulfated fucans and fucosylated chondroitin sulfates are the most abundant non-mammalian sulfated polysaccharides in marine ecosystems, prevalent in algae and marine invertebrates. From the sea cucumber *Holothuria floridana* (Hf), two distinct sulfated polysaccharides were isolated: HfSF and HfFucCS. HfSF, identified as a sulfated fucan, possesses a repeating structure of [→3)-α-Fuc2,4S-(1→3)-α-Fuc-(1→3)-α-Fuc2S-(1→3)-α-Fuc2S-(1→]n. HfFucCS, a fucosylated chondroitin sulfate, is defined by [→3)-β-GalNAc4,6S-(1→4)-β-GlcA-[(3→1)Y]-(1→]n, where Y = αFuc2,4S (45%), α-Fuc3,4S (35%), or α-Fuc4S (20%) [33]. Another sulfated fucan, LvSF, extracted from the sea urchin *Lytechinus variegatus*, exhibits a repeating sequence of [→3)-α-Fuc2,4S-(1→3)-α-Fuc2S-(1→3)-α-Fuc2S-(1→3)-α-Fuc4S-(1→]n [34]. Furthermore, PpFucCS, a fucosylated chondroitin sulfate from the sea cucumber *Pentacta pygmaea*, presents a structure of [→3)-β-GalNAcX(1→4)-β-GlcA-[(3→1)Y]-(1→]n, where X = 4S (80%), 6S (10%), or non-sulfated (10%), and Y = α-Fuc2,4S (40%), α-Fuc2,4S(1→4)-α-Fuc (30%), or α-Fuc4S (30%) [35] (Figure 1). These structures highlight the diverse and complex nature of marine-derived sulfated polysaccharides, emphasizing their potential in biomedical and biotechnological applications.

This study examined the inhibition of *M. pneumoniae* proteins’ interactions with immobilized heparin by these marine-derived polysaccharides via solution/surface competition SPR assays. Solutions of P30 (500 nM) and P1-C (100 nM) were individually combined with each marine sourced sulfated glycan at 10 μg/mL. The results of the solution competition SPR experiments are shown in Figure 5. Heparin achieved a 79% inhibition of P30 protein’s binding to surface-immobilized heparin. LvSF and HfFucCS presented comparable inhibitory effects on the binding of P30 to immobilized heparin, at 66% and 80%, respectively. Furthermore, HfSF and PpFucCS exhibited more pronounced inhibitory activities, effectively preventing P30’s binding with efficiencies of 87% and 94%, respectively. Similarly, for the P1-C protein, heparin exhibited a 79% inhibition of its binding to the immobilized counterpart. LvSF displayed significant inhibitory activity against P1-C binding, with an inhibition rate of 61%. Notably, HfFucCS, HfSF, and PpFucCS demonstrated substantial inhibitory effects on P1-C’s binding, achieving inhibition rates of 89%, 97%, and 100%, respectively. These results underscore the potent inhibitory capabilities of these marine-derived sulfated polysaccharides against protein–heparin interactions, highlighting their potential therapeutic applications.

The study revealed that all the naturally derived marine sulfated glycans (IbSF, IbFucCS, HfSF, HfFucCS, PpFucCS, and LvSF) possess the ability to inhibit the interactions between *M. pneumoniae* proteins and surface-immobilized heparin, as detailed in Table 2. In contrast, their chemically desulfated counterparts, desIbSF and desIbFucCS, showed a marked decrease in inhibitory effectiveness against the binding of both P1-C and P30 proteins to immobilized heparin. These findings underscore the pivotal role of sulfation in the bioactivity of marine sulfated glycans, highlighting its essential contribution to their inhibitory potential. IbSF, LvSF, and HfSF share a basic fucan tetrasaccharide repeating unit yet displaying unique sulfation patterns. Specifically, LvSF, characterized by a higher degree of sulfation, consists of pentasulfated tetrasaccharide motifs, whereas both IbSF and HfSF feature tetrasulfated tetrasaccharide units. The sequence of inhibitory efficacy among these glycans ranges from strongest to weakest as follows: IbSF, HfSF, and then LvSF, with IbSF and HfSF exhibiting greater potency than LvSF. This observation allowed us to conclude that not only sulfation content but also sulfation pattern play a key role in the interaction and inhibition processes studied here by SPR. For further analysis, PpFucCS, IbSF, and HfSF, recognized as the marine-derived sulfated glycans with the strongest inhibitory impact on P30, were chosen for a solution competition dose–response study specifically focusing on P30. Each demonstrated notable inhibitory potential, with PpFucCS distinguishing itself through more pronounced inhibitory effects, as evidenced by the half-maximal inhibitory concentrations (IC_50_) values of 7.0 ng/mL. IbSF and HfSF showed comparable levels of inhibition, underscoring the significant bioactivity of these glycans against the P30 protein. IbFucCS, HfFucCS, and PpFucCS represent a trio of marine-derived fucosylated chondroitin sulfates, each with distinct branching and disulfation levels of fucoses at 96%, 80%, and 70%, respectively. Intriguingly, PpFucCS demonstrated superior inhibitory efficacy against *M. pneumoniae* proteins as compared to the slightly less potent IbFucCS and HfFucCS. The main difference between PpFucCS vs. IbFucCS/HfFucCS is the monosulfated difucosylated motif in PpFucCS (Figure 1). This motif might play a role in enhancing the inhibition of this marine GAG. Regardless of the distinct structures and potencies, our observations led to the selection of all three fucosylated chondroitin sulfates for further solution competition dose–response analysis focusing on the P1-C protein. In the competitive SPR analysis, a series of concentrations of these three marine-sourced glycans were individually mixed with 100 nM P1-C protein. The IC_50_ values, indicative of a 50% reduction in resonance units (RUs), are presented in Table 3. Compared to heparin, the fucosylated chondroitin sulfates exhibited stronger inhibitory effects, with estimated IC_50_ values ranging between approximately 9.3 and 14.1 ng/mL, further substantiating the above findings. These results suggest that the specific pattern of sulfation plays a more critical role in modulating interactions with *M. pneumoniae* proteins than merely the overall degree of sulfation. Our previous research has demonstrated that marine-derived sulfated glycans, particularly Rhamnan sulfate, effectively inhibit SARS-CoV-2 viral entry, showcasing significant antiviral potential in vivo [36]. Given the broad mechanism of action involving the inhibition of entry processes, it is reasonable to hypothesize that these glycans could similarly impede *M. pneumoniae* infections. Future in vivo studies are crucial to assess the potential of marine-sourced sulfated glycans as novel therapies for respiratory infections.

## 3. Materials and Methods

### 3.1. Materials

Porcine intestinal heparin, with an average molecular weight (MW) of 15 kDa and a polydispersity of 1.4, was sourced from Celsus Laboratories (Cincinnati, OH, USA). 6-*O*-desulfated heparin(6-DeS heparin, MW of 13 kDa) was provided by Dr. Wang from the University of South Florida.2-*O*-desulfated IdoA heparin (2-DeS heparin, MW of 13 kDa) and *N*-desulfated heparin (N-DeS heparin, MW of 14 kDa) were synthesized in our laboratory as per the methods described by Yates et al. Heparin oligosaccharides with varying degrees of polymerization (dp4, dp6, dp8, dp10, dp12, dp14, dp16, and dp18) were acquired from Iduron (Manchester, UK). Marine-derived sulfated glycans and their derivatives including IbSF (~100 kDa), desIbSF, IbFucCS (70–80 kDa), desIbFucCS, PpFucCS (~60 kDa), LvSF (~100 kDa), HfSF (~100 kDa), and HfFucCS (60–100 kDa) were isolated from sea cucumbers *Isostichopus badionotus*, *Holothuria floridana*, and *Pentacta pygmaea*, as well as from the sea urchin *Lytechinus variegatus*, in Dr. Pomin’s laboratory at the University of Mississippi [37]. *M. pneumoniae* proteins P1-C and P30 were procured from Prospec Bio (East Brunswick, NJ, USA). Sensor streptavidin (SA) chips used in the studies were purchased from Cytiva (Uppsala, Sweden). SPR measurements were conducted on a T200 SPR instrument (Uppsala, Sweden)using T200evaluation software (version 3.2)for SPR data processing.

### 3.2. Preparation of Heparin Biochip

Biotinylated heparin was synthesized through a procedure that began with dissolving 1 mg of heparin and 1 mg of amine-PEG_3_-Biotin (Thermo Scientific, Waltham, MA, USA) in 200 μL of water. This solution was then enhanced by the addition of 5 mg NaCNBH_3_ and incubated at 70 °C for a 24 h period. Subsequently, an additional 5 mg of NaCNBH_3_ was introduced, and the reaction was extended for another 24 h. Upon completion of the reaction, the mixture was desalted via a spin column (3000 molecular weight cut-off), and the resulting biotinylated heparin was freeze-dried, preparing it for use in chip formation. For the SPR study, the preparation of a heparin SA chip involved injecting a 20 μL solution of the biotinylated heparin (0.1 mg/mL) in HBS-EP+ buffer (0.01 M HEPES pH 7.4, 0.15 M NaCl, 3 mM EDTA, 0.05% *v*/*v* Surfactant P20) (Cytiva, Uppsala, Sweden) across flow cells 2, 3, and 4 of the SA chips at a flow rate of 10 μL/min. Biotin alone was immobilized on flow cell 1, serving as the control channel. 

### 3.3. Binding Kinetics and Affinity Studies of the Interaction between Heparin and the M. pneumoniae Proteins

The *M. pneumoniae* proteins P1-C and P30 were diluted in HBS-EP+ buffer (pH 7.4) for SPR analysis. Different dilutions of these proteins were injected at a flow rate of 30 μL/min across the sensor surface of the SPR chip. Following each protein injection, the HBS-EP+ buffer was flowed over the sensor surface to allow for dissociation of the protein from the chip for 180 s. To regenerate the SPR chip for subsequent measurements, a 30 μL volume of 2 M NaCl solution was injected. The response was continuously monitored as a sensorgram at 25 °C.

### 3.4. Inhibition Activity of the Sulfated Glycans and Marine Sulfated Glycans on Heparin- M. pneumoniae Protein Interactions

To assess the inhibition of the interaction between *M. pneumoniae* proteins and heparin, a solution containing *M. pneumoniae* proteins was prepared by pre-mixing it with various glycans in HBS-EP+ buffer (pH 7.4). This mixture was then introduced over the heparin-immobilized chip surface at a flow rate of 30 μL/min. Subsequently, a 30 μL volume of 2 M NaCl was injected to facilitate the regeneration of the sensor surface, ensuring its readiness for subsequent experiments. Control experiments were conducted using *M. pneumoniae* proteins to confirm that the sensor surface was fully regenerated between measurements. 

### 3.5. Statistical Analysis

Statistical analyses in this study were conducted using one-way ANOVA followed by the Tukey test for multiple comparisons. The Graph Pad Prism 9 software was utilized for all statistical evaluations. The levels of statistical significance were defined as follows: ns for *p* > 0.05, * for *p* < 0.05, ** for *p* < 0.01, *** for *p* < 0.001, and **** for *p* < 0.0001.

## 4. Conclusions

This study elucidated the strong binding affinity of *M. pneumoniae* proteins for heparin and explored the interaction between these proteins and marine-sourced sulfated glycans of diverse and unique structures. Through solution competition assays involving surface-immobilized heparin and heparin oligosaccharides spanning dp4 to dp18, it was discovered that the binding affinity did not correlate with oligomer length. Comparative analysis of desulfated heparin variants highlighted a diminished binding affinity relative to native heparin, underscoring the crucial role of negative charge density in facilitating these interactions. Further, solution competition assays with eight distinct marine-derived sulfated glycans (IbSF, desIbSF, IbFucCS, desIbFucCS, HfSF, HfFucCS, PpFucCS, and LvSF) against surface-immobilized heparin revealed that natural marine sulfated glycans exhibit pronounced inhibitory activity on the binding of *M. pneumoniae* proteins P30 and P1-C. Contrastingly, fully desulfated compounds desIbSF and desIbFucCS demonstrated a lack of inhibitory effect, indicating a dependency of binding on sulfation degree and pattern. This comprehensive library of marine-sourced natural sulfated glycans showcases promising therapeutic potential for mitigating *M. pneumoniae* infections. Despite the apparent inhibitory efficacy of these glycans against the viral proteins in SPR assays with heparin-immobilized surfaces, a direct correlation between the glycans’ structural nuances and their inhibitory capabilities remains elusive. Future research should pivot towards delineating the structure–activity relationships and bioavailability of these sulfated glycans to harness their full therapeutic potential. 

## Figures and Tables

**Figure 1 marinedrugs-22-00232-f001:**
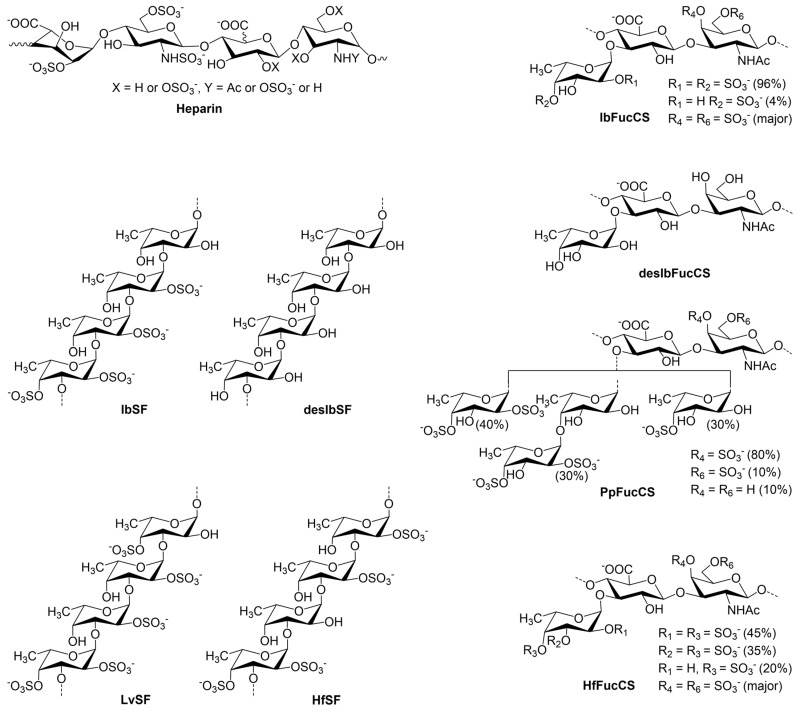
Chemical structures of heparin and the various marine sulfated glycans.

**Figure 2 marinedrugs-22-00232-f002:**
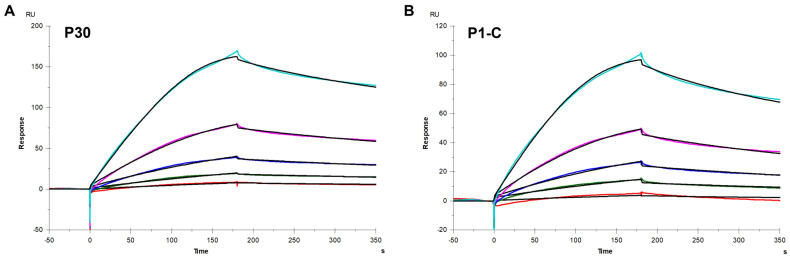
SPR sensorgrams of *M. pneumoniae* proteins binding with heparin. (**A**) SPR sensorgrams of protein P30 binding with heparin. Concentrations of P30 were 500, 250, 125, 62.5, and 31.3 nM (from top to bottom, respectively). (**B**) SPR sensorgrams of P1-C protein binding with heparin. Concentrations of P1-C were 100, 50, 25, 12.5, and 6.25 nM (from top to bottom, respectively). The black curves are the fits using models from T200 Evaluate software (v3.2).

**Figure 3 marinedrugs-22-00232-f003:**
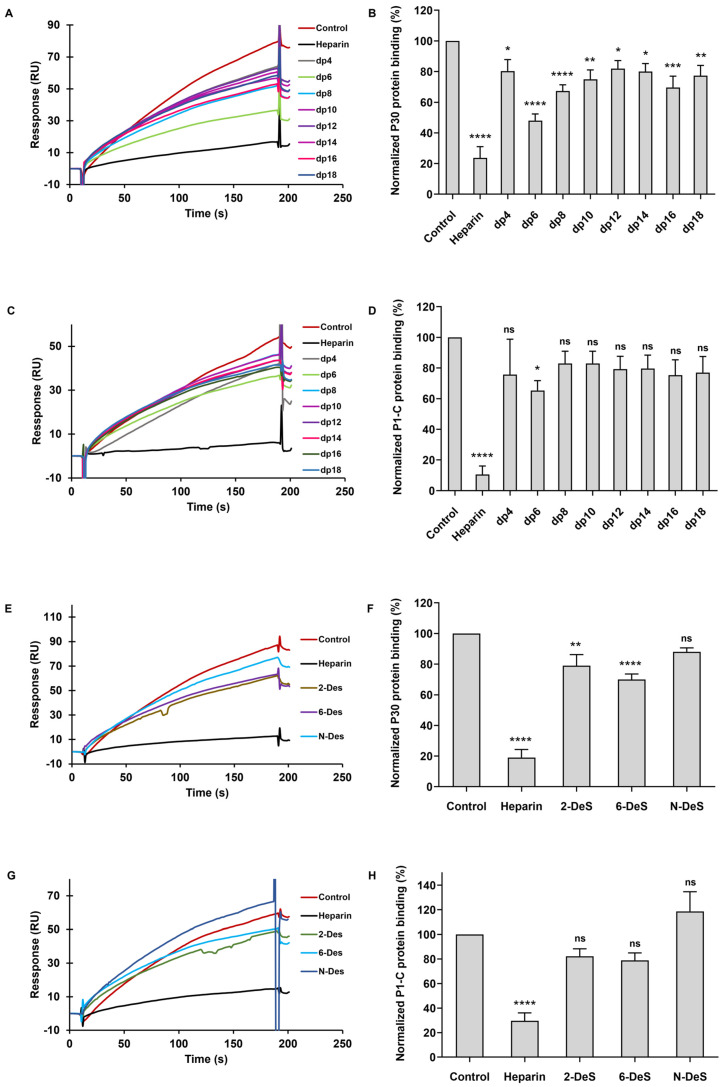
*M. pneumoniae* proteins–heparin interaction inhibited by heparin oligosaccharides and desulfated heparins using solution competition. (**A**) SPR sensorgrams of protein P30–heparin interaction competing with different heparin oligosaccharides. (**B**) Bar graphs of protein P30 binding preference to surface heparin by competing with different heparin oligosaccharides. (**C**) SPR sensorgrams of protein P1-C–heparin interaction competing with different heparin oligosaccharides. (**D**) Bar graphs of protein P1-C binding preference to surface heparin by competing with different heparin oligosaccharides. (**E**) SPR sensorgrams of protein P30–heparin interaction competing with different desulfated heparins. (**F**) Bar graphs of protein P30 binding preference to surface heparin by competing with different desulfated heparins. (**G**) SPR sensorgrams of protein P1-C–heparin interaction competing with different desulfated heparins. (**H**) Bar graphs of protein P1-C binding preference to surface heparin by competing with different desulfated heparins. Data are shown as mean ± SD and are analyzed using a one-way ANOVA/Tukey test. Significance is defined as *p* > 0.05 (ns), *p* < 0.05 (*), *p* < 0.01 (**), *p* < 0.001 (***), *p* < 0.0001 (****).

**Figure 4 marinedrugs-22-00232-f004:**
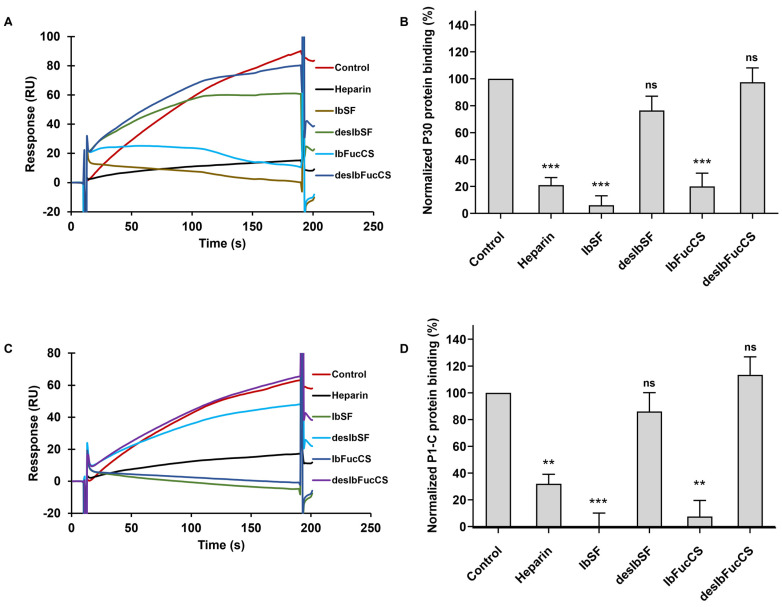
Solution competition between heparin and Ib glycans. (**A**) SPR sensorgrams of *M. pneumoniae* protein P30–heparin interaction competing with IbSF, IbFucCS, desIbSF, and desIbFucCS. (**B**) Bar of normalized protein P30 binding preference to surface heparin by competing with different Ib glycans. (**C**) SPR sensorgrams of M. pneumoniae protein P1-C–heparin interaction competing with IbSF, IbFucCS, desIbSF, and desIbFucCS. (**D**) Bar of normalized protein P1-C binding preference to surface heparin by competing with different Ib glycans. Data are shown as mean ± SD and are analyzed using a one-way ANOVA/Tukey test. Significance is defined as *p* > 0.05 (ns), *p* < 0.01 (**), *p* < 0.001 (***).

**Figure 5 marinedrugs-22-00232-f005:**
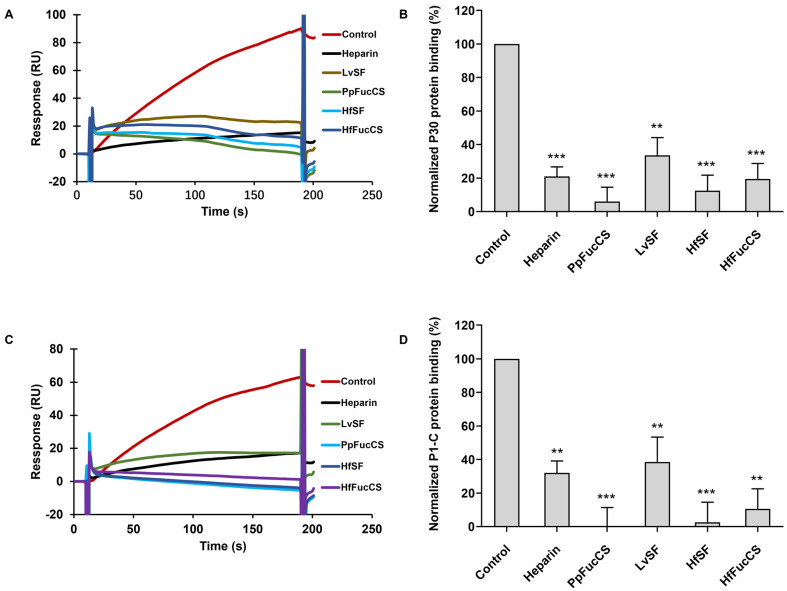
Solution competition between heparin and marine-soured sulfated glycans. (**A**) SPR sensorgrams of protein P30–heparin interaction competing with sulfated glycans derived from marine invertebrates. (**B**) Bar graphs of normalized protein P30 binding preference to surface heparin by competing with different sulfated glycans derived from marine invertebrates. (**C**) SPR sensorgrams of protein P1-C–heparin interaction competing with sulfated glycans derived from marine invertebrates. (**D**) Bar graphs of normalized protein P1-C binding preference to surface heparin by competing with different sulfated glycans derived from marine invertebrates. Data are shown as mean ± SD and are analyzed using a one-way ANOVA/Tukey test. Significance is defined *p* < 0.01 (**), *p* < 0.001 (***).

**Table 1 marinedrugs-22-00232-t001:** Kinetic data of *M. pneumoniae* proteins binding with heparin surface.

	ka (M^−1^ S^−1^)	k_d_ (S^−1^)	*K_D_* (M)
P30	1.02 × 10^5^ (±1.20 × 10^3^) *	2.76 × 10^−3^ (±1.50 × 10^−5^) *	8.03 × 10^−8^ (±3.76 × 10^−8^) **
P1-C	4.46 × 10^5^ (±7.20 × 10^3^) *	3.88 × 10^−3^ (±3.00 × 10^−5^) *	1.64 × 10^−8^ (±5.68 × 10^−9^) **

* The data with (±) in parentheses represent the standard deviation (SD) obtained from the global fitting of five injections. ** SD based on triplicate measurements.

**Table 2 marinedrugs-22-00232-t002:** Summary of solution competition between heparin and eight marine-derived glycans binding to *M. pneumoniae* proteins.

	Control	Heparin	IbSF	desIbSF	IbFucCS	desIbFucCS	HfSF	HfFucCS	PpFucCS	LvSF
Inhibition of P30 binding	0%	79%	94%	23%	80%	2%	87%	80%	94%	66%
Inhibition of P1-C binding	0%	68%	100%	14%	92%	−13%	97%	89%	100%	61%

**Table 3 marinedrugs-22-00232-t003:** Summary of IC_50_ measurement between heparin and marine-derived glycans binding to P30 protein and P1-C protein.

P30 IC_50_ (ng/mL)	**Heparin**	**PpFucCS**	**HfSF**	**IbSF**
	1056.5	7.0	38.6	61.7
P1-C IC_50_ (ng/mL)	**Heparin**	**IbFucCS**	**PpFucCS**	**HfFucCS**
	998.0	14.1	9.3	9.8

## Data Availability

Data available on request from the authors.

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
