# Peer review of "Marine-Derived Sulfated Glycans Inhibit the Interaction of Heparin with Adhesion Proteins of Mycoplasma pneumoniae"

_marinedrugs, 2024, doi:10.3390/md22050232_

Round 1

Reviewer 1 Report

Comments and Suggestions for Authors

This work investigates different sulfated GAGs, specifically heparin, and their ability to inhibit binding of proteins from mycoplasma pneumoniae to host cell antigens. The P1-C and P30 proteins of mycoplasma pneumoniae were the focus of this work due to their importance in adhesion. This work focused primarily on marine derived GAG structures, including fucans and fucosylated CS samples. The results highlighted the key role that sulfation plays in adhesion by testing varying chain lengths, and degrees of sulfation in different structures. The researchers showed that location of sulfation and degrees of polymerization did not greatly effect binding, however extent of sulfation did. I believe this work is sound, and shows a well thought-out approach to testing various GAG structures against mycoplasma pneumoniae infection. This article should be accepted for publication after addressing the following minor issues:

1. Clarification is needed early on as to why marine-derived GAGs were looked at for this study. I might just be missing this since I am not a marine-scientist, but I could not find a straight forward reason why these were the focus.

2. The library of GAGs included heparin, and the marine-derived polysaccharides, which were mainly fucosylated CS and fucans. Was porcine CS ever tested as a direct comparison to the Hp and fucosylated CS?

3. In results section 2.2 the sentence "GAGs form an essential class of macromolecules characterized by their long, linear chains of polysaccharides, which are sulfated and bear a negative charge" is a bit confusing. GAGs are long linear chains. Here it sounds more like you are describing PGs? The wording just needs to be adjusted a bit. 

4. The SPR sensorgrams for figures 3 and 4 are a bit hard to read due to the number of samples shown. Is it possible to change the lines to dashed or dotted? Or is it possible to make the thickness of the line less? I understand this may not be possible due to software constraints.

5. when comparing the different dp lengths, you note there are no major difference between chain lengths, and instead the degree of sulfation matters. Were different structures tested for the dp lengths? Meaning was a dp 14 tested with 1, 2 and 3 sulfates per disaccharide? Does a dp14 with 3 sulfates per disaccharide outperform a dp4 with 3 sulfates per disaccharide? Does the extent of sulfation only matter when considering the disaccharide and not the whole glycan?

6. have you thought to test other glycan structures with negative charges besides sulfation? I would be curious to know is sialic acid had the same effect. Alternatively testing a phosphorylated glycan, or a sulfated N-glycan may be interesting. These experiments are not needed for this work, but it would be interesting to hear your thoughts on how those glycan types may compare to those tested.

Comments on the Quality of English Language

1. Two very minor grammar issues to fix- the first is on line 111 "Heparin, due to its structural...." need a comma after "heparin" (as I added here).

2. The second is on line 234, where it states "...SPR experiments are showed in Figure 5" showed should be "shown".

Author Response

This work investigates different sulfated GAGs, specifically heparin, and their ability to inhibit binding of proteins from mycoplasma pneumoniae to host cell antigens. The P1-C and P30 proteins of mycoplasma pneumoniae were the focus of this work due to their importance in adhesion. This work focused primarily on marine derived GAG structures, including fucans and fucosylated CS samples. The results highlighted the key role that sulfation plays in adhesion by testing varying chain lengths, and degrees of sulfation in different structures. The researchers showed that location of sulfation and degrees of polymerization did not greatly effect binding, however extent of sulfation did. I believe this work is sound, and shows a well thought-out approach to testing various GAG structures against mycoplasma pneumoniae infection. This article should be accepted for publication after addressing the following minor issues:

1. Clarification is needed early on as to why marine-derived GAGs were looked at for this study. I might just be missing this since I am not a marine-scientist, but I could not find a straight forward reason why these were the focus.

Response:  Thank you for your valuable comments. We have provided rational   to use marine-derived sulfated glycans in the Introduction of our revised manuscript. This new section provides a detailed background, highlighting the pharmacological potential of natural products from marine sources. “Furthermore, natural sulfated glycans from marine organisms exhibit unique sulfation patterns and molecular structures, distinct from those found in terrestrial sources. These structural variances confer diverse biological properties and interactions, making marine-derived glycans a valuable resource for novel biomedical applications. Marine sulfated polysaccharides have been demonstrated to possess potential anticoagulant, antitumor, and antiviral activities. With increasing emphasis on sustainability, marine sources provide a viable alternative to terrestrial sources for harvesting biologically active compounds, and they also offer promising targets for exploring novel anti-adherence therapies.” (Vasconcelos, A.; Pomin, V. The Sea as a Rich Source of Structurally Unique Glycosaminoglycans and Mimetics. Microorganisms 2017, 5, 51, doi:10.3390/microorganisms5030051.; Glycans in Diseases and Therapeutics; Pavão, M.S.G., Ed.; Springer Berlin Heidelberg: Berlin, Heidelberg, 2011; ISBN 978-3-642-16832-1.) 

2. The library of GAGs included heparin, and the marine-derived polysaccharides, which were mainly fucosylated CS and fucans. Was porcine CS ever tested as a direct comparison to the Hp and fucosylated CS?

Response:  Thank you for your insightful comment regarding the inclusion of CS in our comparison of GAGs. Based on your suggestion, we expanded our study to evaluate the inhibitory potential of different CS types (CSA, CSB, CSD, and CSE) on the interaction between heparin and M. pneumoniae proteins P30 and P1-C. We utilized solution/surface competition SPR assays to assess this interaction. As shown in Figure R1A and R1B in our results, all types of CS exhibited very low inhibitory efficacy in competing with heparin for binding to the M. pneumoniae proteins. These findings suggest that while CS can interact with these proteins, their inhibitory potential is significantly less pronounced than that of heparin. This result reinforces the specificity and stronger binding affinity of heparin in this biological context, which may be crucial for future therapeutic applications targeting M. pneumoniae infections.

Figure R1: M. pneumoniae proteins-heparin interaction inhibited by CSA, CSB, CSD and CDE using solution competition SPR. (A) Bar graphs of protein P30 binding preference to surface heparin by competing with different CSs, the concentration used was 250 nM for P30 protein and 1000 nM for the CSs (B) Bar graphs of protein P1-C binding preference to surface heparin by competing with different CSs, the concentration was 50 nM for P1-C protein and 1000 nM for the CSs.

3. In results section 2.2 the sentence "GAGs form an essential class of macromolecules characterized by their long, linear chains of polysaccharides, which are sulfated and bear a negative charge" is a bit confusing. GAGs are long linear chains. Here it sounds more like you are describing PGs? The wording just needs to be adjusted a bit.

Response: We have revised the sentence to reflect the characteristics of GAGs more accurately: "GAGs are an essential class of macromolecules characterized by their long, linear chains of negatively charged polysaccharides."

4. The SPR sensorgrams for figures 3 and 4 are a bit hard to read due to the number of samples shown. Is it possible to change the lines to dashed or dotted? Or is it possible to make the thickness of the line less? I understand this may not be possible due to software constraints.

Response: We have reduced the line thickness and adjusted the line colors in Figures 3 and 4 to enhance contrast and improve clarity.  

5. When comparing the different dp lengths, you note there are no major difference between chain lengths, and instead the degree of sulfation matters. Were different structures tested for the dp lengths? Meaning was a dp 14 tested with 1, 2 and 3 sulfates per disaccharide? Does a dp14 with 3 sulfates per disaccharide outperform a dp4 with 3 sulfates per disaccharide? Does the extent of sulfation only matter when considering the disaccharide and not the whole glycan?

Response: In our study, we compared oligosaccharides with varying degrees of polymerization, featuring different numbers of disaccharide repeat units. While longer chains naturally have more sulfate groups, they did not necessarily outperform shorter ones. This suggests that while the degree of sulfation is crucial, its effect is not solely dependent on the disaccharide unit but rather on the entire glycan structure. Further research will be required to understand the nuances of this relationship more comprehensively.

 6. Have you thought to test other glycan structures with negative charges besides sulfation? I would be curious to know is sialic acid had the same effect. Alternatively testing a phosphorylated glycan, or a sulfated N-glycan may be interesting. These experiments are not needed for this work, but it would be interesting to hear your thoughts on how those glycan types may compare to those tested.

Response: Thank you for your insightful suggestions about incorporating other negatively charged glycan structures into our research. Indeed, exploring the effects of sialic acid, phosphorylated glycans, and sulfated N-glycans offers a compelling direction for future studies. These glycans, each with unique charge profiles and structural characteristics, could provide additional insights into glycan-protein interactions. Our referenced study, "Distinct Mycoplasma pneumoniae Interactions with Sulfated and Sialylated Receptors (https://doi.org/10.1128/IAI.00392-20)," highlights the unique roles of sialic acids in microbial adhesion and pathogenesis. It shows that while M. pneumoniae effectively binds to sialylated receptors, which are crucial for the pathogen's motility and invasive activities, it interacts differently with sulfated receptors, which do not support gliding motility and may play roles in initial attachment and immune evasion. Additional research such as that by Song et al. (https://doi.org/10.1074/jbc.M111.274217). has explored how modified sialic acids affect interactions with proteins and viruses, potentially illuminating unique binding behaviors. Studies on phosphorylated and sulfated N-glycans, such as those by Khoo and Yu (https://doi.org/10.1016/S0076-6879(10)78001-0) and Yamada et al. (https://doi.org/10.1021/acs.analchem.8b00714), have highlighted their importance in contexts like cancer and inflammation, and their roles in disease diagnostics. These studies demonstrate how modifications in glycan structures can significantly influence their biological functions and interactions, providing insights into potential therapeutic targets and diagnostic markers.

While these studies were not within the scope of our current work, they align well with our long-term objectives to understand and leverage glycan dynamics in disease processes. We believe that such comparative analyses could elucidate additional mechanisms of action and potentially uncover broader applications in biomedicine.

Comments on the Quality of English Language

Two very minor grammar issues to fix- the first is on line 111 "Heparin, due to its structural...." need a comma after "heparin" (as I added here).  the second is on line 234, where it states "...SPR experiments are showed in Figure 5" showed should be "shown".

Response: The minor grammar issues have been fixed in our revised manuscript.

Reviewer 2 Report

Comments and Suggestions for Authors

The manuscript by Jiyuan Yang. et al. indicated that M. pneumoniae adhesion proteins have strong binding affinity for heparin and this binding affinity between them could be inhibited by a variety of marine-derived sulfated glycans. In addition, the extent of binding inhibition is independent of glycan chain length. Furthermore, the author also showed that sulfation acts as a key structural feature for the interaction of marine-derived sulfated glycans with M. pneumoniae adhesion proteins. While I found these data is interesting, I still have a few comments and questions.

Strength

It is a valuable work for showing the inhibitory effect of marine-derived sulfated glycans on the interaction of heparin with M. pneumoniae adhesion proteins.

Limitation

The author prove the inhibitory effect of sulfated glycans on the interaction of heparin with M. pneumoniae adhesion proteins is closely correlated to the specific pattern of sulfation. However, it is necessary to elucidate inhibitory mechanism of the specific sulfation pattern on the interaction of heparin with M. pneumoniae adhesion proteins and more experiments are needed.

1. In addition to glycan chain length and sulfation degree, the biological activity of sulfated glycans is also determined by other factors, such as disaccharide composition, relative molecular mass, charge, and so on. How the author avoid the above factors?

2. Why the author selected marine as the primary source of sulfated glycans that exert inhibitory effect?

3. In line 235-245, LvSF, HfFucCS, HfSF, and PpFucCS all have relevant potent inhibitory capabilities against protein-heparin interactions, notably, HfFucCS has a stronger inhibitory effect on P1-C's binding than P30's binding, if it has a certain structural specificity for the P1-C protein? I look forward to seeing you add this section to the discussion.

4. What is the potential inhibitory mechanism of sulfated glycans on the interaction of heparin with M. pneumoniae adhesion proteins? Should add some experiments to further explore this?

5. When looking at the interesting in vitro data about the inhibitory effect of marine-derived sulfated glycans on the interaction of heparin with M. pneumoniae adhesion proteins, I just wondering that the authors have a plan to study the application value and therapeutic potential of these marine-derived sulfated glycans on the human or animals M. pneumoniae infection model? I totally understand that it is not an easy experiment to do but, at some point, might be very valuable to validate the therapeutic effect and functional mechanisms. Or, at least, the authors should add some discussion to mention it through cite some papers.

Comments on the Quality of English Language

Minor editing of English language required

Author Response

The manuscript by Jiyuan Yang. et al. indicated that M. pneumoniae adhesion proteins have strong binding affinity for heparin and this binding affinity between them could be inhibited by a variety of marine-derived sulfated glycans. In addition, the extent of binding inhibition is independent of glycan chain length. Furthermore, the author also showed that sulfation acts as a key structural feature for the interaction of marine-derived sulfated glycans with M. pneumoniae adhesion proteins. While I found these data is interesting, I still have a few comments and questions.

Strength

It is a valuable work for showing the inhibitory effect of marine-derived sulfated glycans on the interaction of heparin with M. pneumoniae adhesion proteins.

Limitation

The author prove the inhibitory effect of sulfated glycans on the interaction of heparin with M. pneumoniae adhesion proteins is closely correlated to the specific pattern of sulfation. However, it is necessary to elucidate inhibitory mechanism of the specific sulfation pattern on the interaction of heparin with M. pneumoniae adhesion proteins and more experiments are needed.

1. In addition to glycan chain length and sulfation degree, the biological activity of sulfated glycans is also determined by other factors, such as disaccharide composition, relative molecular mass, charge, and so on. How the author avoid the above factors?

Response: Thank you for your positive and constructive comments.  It is an important point about the molecular complexities involved in studying the biological activity of sulfated glycans. Indeed, factors such as disaccharide composition, relative molecular mass, charge density, and others can significantly influence the activity of these molecules. In our manuscript, we have discussed the impact of these factors in various sections. For example, in Section 2.1, we used a series of heparin oligosaccharides with varying numbers of disaccharide repeat units to explore the effects of polymerization degree. In the desulfated heparins section, the differences between samples were solely due to variations in sulfation sites, allowing us to discuss the importance of sulfation and the distinct effects of different sulfation sites. These discussions are intended to provide a comprehensive understanding of how each factor contributes to the overall bioactivity of sulfated glycans.

2. Why the author selected marine as the primary source of sulfated glycans that exert inhibitory effect?

Response: We chose marine sources primarily because they are rich in diverse and structurally unique sulfated glycans. These unique structural features potentially lead to different biological interactions and activities, making marine-derived glycans a valuable resource for discovering novel bioactive compounds. And previous research indicates that marine-derived sulfated glycans often exhibit strong biological activities, such as anticoagulant and antiviral effects, which suggested potential for similar activities against microbial adhesion proteins.    Thus, using marine sources provided a promising avenue to explore new and effective inhibitors of pathogen-host interactions.

We have revised the introduction of our manuscript to clearly outline our focus on marine-derived GAGs: “Furthermore, natural sulfated glycans from marine organisms exhibit unique sulfation patterns and molecular structures, distinct from those found in terrestrial sources. These structural variances confer diverse biological properties and interactions, making marine-derived glycans a valuable resource for novel biomedical applications. Marine sulfated polysaccharides have been demonstrated to possess potential anticoagulant, antitumor, and antiviral activities. With increasing emphasis on sustainability, marine sources provide a viable alternative to terrestrial sources for harvesting biologically active compounds, and they also offer promising targets for exploring novel anti-adherence therapies.”

3. In line 235-245, LvSF, HfFucCS, HfSF, and PpFucCS all have relevant potent inhibitory capabilities against protein-heparin interactions, notably, HfFucCS has a stronger inhibitory effect on P1-C's binding than P30's binding, if it has a certain structural specificity for the P1-C protein? I look forward to seeing you add this section to the discussion.

Response: We appreciate your insightful comments regarding the stronger inhibitory effects of HfFucCS on the binding interactions of P1-C and P30 proteins with heparin. Your suggestion to further explore the structural specificity of HfFucCS for the P1-C protein is particularly valuable. In our study, the difference in inhibition between P1-C and P30 might indeed suggest a specific interaction affinity or structural compatibility of HfFucCS with the P1-C protein. This observation warrants a more detailed structural analysis to understand how the molecular configurations of HfFucCS may preferentially influence its interaction with different proteins.

4. What is the potential inhibitory mechanism of sulfated glycans on the interaction of heparin with M. pneumoniae adhesion proteins? Should add some experiments to further explore this?

Response: We hypothesize that sulfated glycans interfere with these interactions by mimicking structural elements of heparin, thereby competitively inhibiting binding sites on the adhesion proteins. Our competitive SPR assays in the manuscript is a commonly employed experimental method for such demonstrations. Furthermore, in future studies, we could enhance our findings through molecular docking simulations to predict interaction sites between heparin or marine-derived drugs and the proteins. By identifying predicted interaction sites, we can perform mutagenesis of key amino acids followed by subsequent binding experiments to validate our hypothesis. These advanced experiments would provide a more comprehensive understanding of the inhibitory mechanisms of sulfated glycans on these protein interactions.

5. When looking at the interesting in vitro data about the inhibitory effect of marine-derived sulfated glycans on the interaction of heparin with M. pneumoniae adhesion proteins, I just wondering that the authors have a plan to study the application value and therapeutic potential of these marine-derived sulfated glycans on the human or animals M. pneumoniae infection model? I totally understand that it is not an easy experiment to do but, at some point, might be very valuable to validate the therapeutic effect and functional mechanisms. Or, at least, the authors should add some discussion to mention it through cite some papers.

Response: The potential application and therapeutic value of marine-derived sulfated glycans against Mycoplasma pneumoniae in animal or human models is the next area of exploration. As your suggestion, we have added the following content to the 2.4 discussion section of our manuscript: “Our previous research has demonstrated that marine-derived sulfated glycans, particularly rhamnan sulfate, effectively inhibit SARS-CoV-2 viral entry, showcasing significant antiviral potential in vivo. Given the broad mechanism of action involving the inhibition of entry processes, it is reasonable to hypothesize that these glycans could similarly impede M. pneumoniae infections. Future in vivo studies are crucial to assess the potential of marine-sourced sulfated glycans as novel therapies for respiratory infections.” (Song, Y.; Singh, A.; Feroz, M.M.; Xu, S.; Zhang, F.; Jin, W.; Kumar, A.; Azadi, P.; Metzger, D.W.; Linhardt, R.J.; et al. Seaweed-Derived Fucoidans and Rhamnan Sulfates Serve as Potent Anti-SARS-CoV-2 Agents with Potential for Prophylaxis. Carbohydrate Polymers 2024, 337, 122156, doi:10.1016/j.carbpol.2024.122156.)

Reviewer 3 Report

Comments and Suggestions for Authors

Summary

This manuscript describes the assessment of sulphated polysaccharides on inhibiting interaction between mycoplasma adhesin P1 and P30 and heparin, using surface plasmon resonance on a Biacore T200. The authors tested effects of heparin length, sulphation, and a range of marine polysaccharides from sea cucumbers. The basis of all the interaction relies on a biotinylated 15kDa heparin (approx. 24dp assuming 3 sulphates per disaccharide) immobilised on the chip via streptavidin.

Comments

Overall, the scientific idea is sound, but it would have been beneficial if a few other interaction analyses be performed, as all the interactions are only assessed by inhibition towards heparin binding. For example, directly measure the affinity of the polysaccharides towards P1 or P30, by immobilising the inhibitory sulphated polysaccharides instead of heparin or immobilising the P1 and P30 proteins and injecting in the different polysaccharides.

The authors suggest that DP does not affect inhibition, but Fig3A seems to suggest some form of length dependency, if the dp4 data point is considered as an anomaly. Please include more details about the sulphation status of the various lengths, are they all sulphated to the same degree?

2M NaCl was used as the regeneration condition, and de-sulphation reduces the inhibition. Do free sulphates, or other anions such as phosphates and carbonates, do the same?

Please include more details regarding the various marine polysaccharides in terms of the DP, or average DP, used.

 In general the SPR experiments were done Ok though more control experiments  should be conducted for more convincing data.

Two technical comments are proposed: 

1. Biotinylated heparin was synthesized, however, there is no data showing biotin was conjugated actually with heparin. The authors should provide the full SPR  angular spectra of SA-chip and Heparin immobilized chip, and compare the spectral shift to prove the binding of biotinylated heparin on the SPR chip. 

2. 2 M NaCl was used to regenerate the sensor surface. And it was described in the Method (Page 11) that "Control experiments were conducted using M. pneumoniae proteins to confirm that the sensor surface was fully regenerated between measurements". However, there is no data showing the SPR response of the regenerated surface. The authors should show the control data, and provide the full SPR angular spectra of the chip before and after regeneration of sensor surface. 

Author Response

This manuscript describes the assessment of sulphated polysaccharides on inhibiting interaction between mycoplasma adhesin P1 and P30 and heparin, using surface plasmon resonance on a Biacore T200. The authors tested effects of heparin length, sulphation, and a range of marine polysaccharides from sea cucumbers. The basis of all the interaction relies on a biotinylated 15kDa heparin (approx. 24dp assuming 3 sulphates per disaccharide) immobilised on the chip via streptavidin.

Comments

1. Overall, the scientific idea is sound, but it would have been beneficial if a few other interaction analyses be performed, as all the interactions are only assessed by inhibition towards heparin binding. For example, directly measure the affinity of the polysaccharides towards P1 or P30, by immobilising the inhibitory sulphated polysaccharides instead of heparin or immobilising the P1 and P30 proteins and injecting in the different polysaccharides.

Response: Thank you for your positive and constructive feedback regarding the interaction analyses in our study.   You've raised an excellent point about the need to diversify the methods used to assess the interactions between marine-derived sulfated glycans and Mycoplasma pneumoniae adhesion proteins P1 and P30.  We prefer using heparin-immobilized SPR chip in our interaction study since it is much more stable/robust and the data is more comparable than protein-immobilized chips.

2. The authors suggest that DP does not affect inhibition, but Fig3A seems to suggest some form of length dependency, if the dp4 data point is considered as an anomaly. Please include more details about the sulphation status of the various lengths, are they all sulphated to the same degree?

Response: Thank you for your observations and questions regarding Figure 3A and the discussion about the influence of degree of polymerization on inhibition effectiveness in our study. We evaluated oligosaccharides with varying degrees of polymerization, featuring different numbers of disaccharide repeat units. Each of these oligosaccharides had a similar degree of sulfation per disaccharide repeat unit. Although longer chains naturally contain more sulfate groups, they did not necessarily exhibit superior performance compared to shorter ones. This observation suggests that while the degree of sulfation is crucial, its impact is not solely dependent on the disaccharide unit but rather on the entire glycan structure. Further research is needed to understand the nuances of this relationship more comprehensively.

3. 2M NaCl was used as the regeneration condition, and de-sulphation reduces the inhibition. Do free sulphates, or other anions such as phosphates and carbonates, do the same?

Response: Thank you for your insightful question regarding the use of anions as regeneration agents in SPR experiments.  In SPR, selecting the appropriate regeneration agent is crucial as it directly affects the reusability of the sensor chip and the reproducibility of the experimental results.  Salt solutions are commonly used as regeneration agents to neutralize the electrostatic interactions between molecules.  Using anions such as sulfates, phosphates, or carbonates is theoretically feasible, particularly when the interactions between target molecules are predominantly electrostatic.  These anions can effectively neutralize or disrupt the electrostatic interactions between proteins and other biomolecules, thereby dissociating the formed complexes.  Although these anions can serve as regeneration agents, careful selection and optimization of their concentration and conditions are necessary in practical applications to ensure they can effectively dissociate specific molecular complexes.

4. Please include more details regarding the various marine polysaccharides in terms of the DP, or average DP, used.

Response: We have revised the 3.1. materials section of our manuscript to include detailed information on the molecular weights of the marine polysaccharides used in our studies. We have also cited relevant literature to provide a robust context for these values, ensuring clarity and rigor in our experimental descriptions. “Marine-derived sulfated glycans and their derivatives including IbSF (~100 kDa), desIbSF, IbFucCS (70-80 kDa), desIbFucCS, PpFucCS (~60 kDa), LvSF (~100 kDa), HfSF (~100 kDa), HfFucCS (60-100 kDa) were isolated from sea cucumbers Isostichopus badionotus, Holothuria floridana, and Pentacta pygmaea, as well as from the sea urchin Lytechinus variegatus, in Dr. Pomin’s laboratory at the University of Mississippi.” (Dwivedi, R.; Sharma, P.; Farrag, M.; Kim, S.B.; Fassero, L.A.; Tandon, R.; Pomin, V.H. Inhibition of SARS-CoV-2 Wild-Type (Wuhan-Hu-1) and Delta (B.1.617.2) Strains by Marine Sulfated Glycans. Glycobiology 2022, cwac042, doi:10.1093/glycob/cwac042)

5. In general the SPR experiments were done Ok though more control experiments should be conducted for more convincing data.

Response: Thank you for your feedback on our SPR experiments and the suggestion to conduct more control experiments for robust data validation.  Moving forward, we plan to incorporate in vitro cell viability assay to further corroborate our data and provide deeper insights into the biological relevance of our observations.

6. Two technical comments are proposed: Biotinylated heparin was synthesized, however, there is no data showing biotin was conjugated actually with heparin. The authors should provide the full SPR angular spectra of SA-chip and Heparin immobilized chip, and compare the spectral shift to prove the binding of biotinylated heparin on the SPR chip.

Response: Thank you for your suggestions.  In our SPR studies, we prepared the heparin SPR chip by injecting a 20 μL solution of biotinylated heparin (0.1 mg/mL) in HBS-EP+ buffer across flow cells 2, 3, and 4 of the SA chips.  For control purposes, biotin alone was immobilized on flow cell 1, serving as the control channel. We have provided a sensorgrams in Figure R2, showing the preparation process of the heparin immobilized chip.  The baseline signal increased in flow cells 2, 3, and 4, after injecting with biotinylated heparin, which demonstrates the successful immobilization.  This data effectively supports our preparation process and the functionality of the biotinylated heparin.

Figure R2. SPR sensorgrams of preparation process of the heparin immobilized chip.

7. 2 M NaCl was used to regenerate the sensor surface. And it was described in the Method (Page 11) that "Control experiments were conducted using M. pneumoniae proteins to confirm that the sensor surface was fully regenerated between measurements". However, there is no data showing the SPR response of the regenerated surface. The authors should show the control data, and provide the full SPR angular spectra of the chip before and after regeneration of sensor surface.

Response: We have included the full SPR sensorgrams before and after each regeneration (see Figure R3) to demonstrate that the baseline remains stable across experiments, thus confirming the effective regeneration of the chip.

Figure R3: SPR sensorgrams of protein P30-heparin interaction competing with sulfated glycans derived from marine invertebrates.

Round 2

Reviewer 2 Report

Comments and Suggestions for Authors

Although the authors provided a reasonable explanation of my concerns, I still feel that the in vivo validation of the authors' work is seriously lacking.

Comments on the Quality of English Language

Acceptable.

Reviewer 3 Report

Comments and Suggestions for Authors

Although the authors refused the extra experiments suggested, extra information was added to the methods and the responses were satisfactory